# Room-Temperature Strengthening, Portevin-Le Chatelier Effect, High-Temperature Tensile Deformation Behavior, and Constitutive Modeling in a Lightweight Mg-Gd-Al-Zn Alloy

**DOI:** 10.3390/ma16041639

**Published:** 2023-02-16

**Authors:** Furong Cao, Huizhen Guo, Nanpan Guo, Shuting Kong, Jinrui Liang

**Affiliations:** 1School of Materials Science and Engineering, Northeastern University, Shenyang 110819, China; 2State Key Laboratory of Rolling and Automation, Northeastern University, Shenyang 110819, China

**Keywords:** Mg-Gd-Al-Zn alloy, microstructure, strengthening, Portevin-Le Chatelier effect, high temperature deformation, constitutive model

## Abstract

To explore room-temperature strengthening and high-temperature ductility, a lightweight novel Mg-1.85Gd-0.64Al-0.62Zn alloy was fabricated by innovative multidirectional forging and a hot-rolling technique. Microstructures and mechanical properties were studied at room and elevated temperatures with an optical microscope, an X-ray diffractometer, and a tensile tester. An ultimate tensile strength of 260 MPa, yield strength of 171 MPa, and elongation of 20.4% were demonstrated at room temperature. The room-temperature strengthening mechanisms were identified by strengthening the model estimation. A type C Portevin-Le Chatelier effect was discovered and elucidated in this alloy. X-ray diffraction analysis revealed that the phase composition is α-Mg solid solution and (Mg, Al)_3_Gd, Al_7_Zn_3_, and Al_2_Gd intermetallic compounds. Examination of the microstructure at elevated temperatures revealed that dynamic recrystallization and dynamic grain growth occur. In particular, it was discovered that bimodal microstructures or incomplete dynamic recrystallization microstructures exist in high-temperature deformation. A maximum quasi-superplasticity of 228.4% was demonstrated in this alloy at 673 K and 5.0 × 10^−4^ s^−1^. Flow stress curves showed that the present alloy exhibits Sotoudeh–Bate curves or a long intermediate strain-hardening stage followed by a strain-softening stage. A modified Zerilli–Armstrong constitutive equation incorporating the number of dislocations was established. The power-law constitutive equation was established to identify the deformation mechanism. Both constitutive models had good predictability. At 673 K and 5.0 × 10^−4^ s^−1^, the stress exponent was 4, and the average deformation activation energy was 104.42 kJ/mol. The number of dislocations inside a grain was 146. This characteristic evidence confirmed that dislocation motion controlled by pipe diffusion dominates the rate-controlling process under this condition.

## 1. Introduction

Due to the requirements of green environmental protection and CO_2_ emissions reduction, lightweight magnesium alloys have captured extensive attention in the materials community. In particular, Mg-Gd-Zn and Mg-Gd-Al-Zn alloys, which belong to Mg-Gd system alloys, have the potential for application in biodegradable materials, 3C electronics, and automobile industries because of the low cost of Al and Zn elements. To date, reports on numerous Mg-Gd-Zn alloys [1,2,3,4] and some Mg-Gd-Al-Zn alloys [5,6,7] are available for the attempt to meet the requirements of commercial demand for strength. Due to the limitation of rolling plasticity of high Gd-content magnesium alloys, high Gd Mg alloy ingot is often rolled into pieces due to edge cracks of rolled plates. Hence, we designed a low Gd-content novel Mg-2Gd-1Al-1Zn alloy in wt%. The purpose of the addition of 2 wt% Gd and 1 wt% Al was to achieve solid solution strengthening with a Mg matrix and to form intermetallic compounds such as (Mg, Al)_3_Gd, as well as Al_2_Gd intermetallic compounds, in order to achieve second-phase strengthening. Meanwhile, the purpose of the addition of 1 wt% Zn was to achieve solid solution strengthening and second-phase strengthening with the formation of (Mg, Zn)_3_Gd and Al_7_Zn_3_ intermetallic compounds. On the other hand, severe plastic deformations such as equal channel angular pressing (ECAP) [8,9], high-pressure torsion (HPT) [10,11], friction stir processing (FSP) [12], and multidirectional forging (MDF) [13] of magnesium alloys has attracted the attention of researchers over the past decades. However, no work has studied the microstructures and mechanical properties at room temperature fabricated by MDF and hot rolling. For this reason, it is necessary to investigate the microstructures and mechanical properties of the Mg-2Gd-1Al-1Zn alloy at room temperature fabricated by MDF and hot rolling.

The strengthening mechanism and the Portevin-Le Chatelier effect at room temperature are two interesting topics in some alloys. On the one hand, a strengthening mechanism was investigated in some magnesium alloys [14,15,16], and some recent relevant reports were documented [17,18,19,20]. Research on the strengthening mechanism is developing towards the quantification of individual strengthening mechanisms and elucidation from atomic mechanisms. On the other hand, the Portevin-Le Chatelier (PLC) effect, a serrated flow phenomenon, has attracted widespread attention in Al-Mg system alloys [21,22], nickel superalloys [23], steel [24], and magnesium-lithium alloys [25] and is developing towards the application of digital-imaging correlation techniques and the elucidation of the underlying mechanism. The aim is to lessen and remove the bad influence of the PLC band on the surface quality of the worked piece. To the best of our knowledge, little information is available investigating the strengthening mechanism and PLC effect in the Mg-2Gd-1Al-1Zn alloy. Hence, it is necessary to study the strengthening mechanism and PLC effect in this alloy.

High-temperature deformation behavior such as hot forming, e.g., hot extrusion, rolling, and forging; hot deformation, e.g., hot compression, tension, and torsion; and superplasticity and creep are important fields. Constitutive modeling and dynamic recrystallization (DRX) are interesting topics, for which the implementation of numerical simulation and the calculation of deformation loads, as well as the understanding of high-temperature deformation processes, are necessary. On the one hand, constitutive modeling can be classified with phenomenological equations, physics-based constitutive equations, and artificial neural network (ANN) models [26]. Due to their higher correlation coefficients and lower absolute relative errors, phenomenological equations, such as the Arrhenius equation [27,28] and the modified Johnson–Cook equation [29], were established in magnesium alloys. According to our survey, little work has been reported on the modified Zerilli–Armstrong (MZA) constitutive equation, and no one has linked the number of dislocations to the MZA constitutive equation. On the other hand, DRX was studied in some magnesium alloys [28,30], but the DRX behavior in the Mg-2Gd-1Al-1Zn alloy remains unknown. However, few reports are available that study the MZA constitutive equation considering the number of dislocations and DRX behavior in the Mg-2Gd-1Al-1Zn alloy utilizing high-temperature tensile tests. In addition, the power-law constitutive equation is especially suitable for the identification of deformation mechanisms at an elevated temperature. Superplasticity [31,32,33,34] and hot deformation behavior [27,28,30] of a Mg-Gd-(Zn) alloy were reported. However, there is no report studying the high-temperature ductility and deformation mechanism in the present Mg-2Gd-1Al-1Zn alloy. Hence, it is necessary to investigate the high-temperature microstructure, ductility, constitutive equation, and DRX behavior.

In this work, our research encompasses several aspects: (i) the fabrication of a new Mg-2Gd-1Al-1Zn alloy; (ii) studying its room temperature and high-temperature microstructure and mechanical properties; (iii) uncovering the strengthening and PLC effect; (iv) the construction of a MZA constitutive equation incorporating the number of dislocations and the construction of a power-law constitutive equation in order to elucidate the high-temperature deformation mechanism in the present alloy.

## 2. Experimental Procedures

### 2.1. Materials Preparation

Mg blocks were melted at 993 K in an electric resistance furnace. The protective gas was argon, and the flux was RJ5 flux. Then, an Mg-30Gd master alloy, Al, and Zn blocks were added to the Mg melt. The temperature was raised to 1023 K until all the blocks were melted. The temperature was decreased to 983 K. The melt was casted into an ingot in a copper mold. The analyzed composition was 1.85 wt. % Gd, 0.64 wt. % Al, 0.62 wt. % Zn, and balanced Mg (hereinafter denoted as GAZ211). Homogenized at 673 K for 16 h, the ingot was milled to remove its surface defects. Then, isothermal multidirectional forging (MDF) was performed, and a schematic diagram has been shown elsewhere [25]. The milled ingot was spark-discharge machined into a cuboid with dimensions of 30 mm × 23 mm × 20 mm. Prior to isothermal MDF, the cuboid was heated at 723 K for 20 min, and then the forging process began. The reductions for the first three MDF passes were 33.3, 34.8, and 32.0%, respectively. The reductions for the fourth, fifth, sixth, seventh, eighth, and ninth MDF passes corresponded to the reductions of the first three passes, respectively. After MDF finished and the MDF billets were heated at 693 K for 1 h, the billets were hot rolled for two passes at 693 K from 20.0 mm thick to 15.3 mm thick, with a pass reduction of 15.0 and 10.0%, respectively. Then, the rolled plates were heated at 693 K for 15 min and hot rolled for another two passes to 13.2 mm, with a pass reduction of 8.5 and 6.0%, respectively. The total reduction was 34.0%. After hot rolling, the rolled plates were annealed at 723 K for 1h. Once the abovementioned hot forming was completed, the workpiece was quenched into water to reserve the hot-forming microstructure.

### 2.2. Tensile Tests at Room Temperature and Elevated Temperatures

Room-temperature tensile samples were prepared by spark-discharge machining along the rolling direction or longitudinal direction with gauge dimensions of 10 mm × 6 mm × 2 mm. The tensile velocity was 3 mm/min. Three samples were used for each condition during tensile tests at room temperature. High-temperature tensile samples were prepared by spark-discharge machining along the rolling direction or longitudinal direction with gauge dimensions of 12 mm × 6 mm × 2 mm. High-temperature tensile tests were conducted on a Shimidazu AG-X Plus universal tensile tester at a temperature range of 573~723 K and with a strain rate range of 1.67 × 10^−2^~1.67 × 10^−4^ s^−1^. Once the tensile deformation finished, the sample was quenched into water to reserve the high-temperature microstructure.

### 2.3. Microstructural Characterization

Samples for optical microscope observation were ground with 400, 600, up to 5000# abrasive papers, polished with a polisher to a mirror-like surface, and etched in a solution of 5 g picric acid +5 mL acetic acid +10 mL distilled water +100 mL ethanol. The optical microstructure observation was conducted on an Olympus DSX500 optical microscope. The grain size was measured by Image-Pro-Plus (IPP) software.

Samples for X-ray diffraction analysis were surface-polished by abrasive papers. A PW3040/60 X-ray diffractometer was used to examine the polished specimens. The phase composition was analyzed using X’Pert High Score Plus software.

## 3. Results

### 3.1. Microstructure and Mechanical Properties of the Present Alloy at Room Temperature

Figure 1 shows the microstructures of the GAZ 211 alloy in different states. As shown in Figure 1a, the homogenized alloy is composed of a white hexagonal close-packed (HCP) structured α-Mg solid solution and black network-like intermetallic compounds. The grain boundary becomes rounded, and element segregation in the grain interior is eliminated due to homogenization. The average grain size is 53 ± 4.1 μm. As shown in Figure 1b, after three-pass isothermal multidirectional forging at 723 K, grains are fragmented and refined to a great extent. The network-like coarse intermetallic compound particles are broken into smaller particles distributed uniformly in the microstructure. The average grain size is 26 ± 2.9 μm. Grain refinement and intermetallic compound break-up are due to external forging. Normal stress is transformed into internal shear stress, and mechanical shear stress fragments and refines the grains. Due to the occurrence of DRX, grain refinement occurs. As shown in Figure 1c, after multidirectional forging at 723 K and hot rolling at 693 K, under applied rolling stress, coarse grains are compressed and fragmented into smaller grains, during which DRX occurs. As a result, grains are refined. The average grain size is 22 ± 2.5 μm. As shown in Figure 1d, after annealing at 723 K for 1 h, uniform equiaxed grains are formed due to static recrystallization. The average grain size is 30 ± 3.1 μm.

Figure 2 presents the X-ray diffraction patterns of the GAZ 211 alloy under different states. The constituent phases under as-cast, multidirectional forging, and hot-rolling states are HCP α-Mg solid solution and (Mg, Al)_3_Gd, Al_7_Zn_3_, and Al_2_Gd intermetallic compounds. With the progress of processing from the as-cast state to multidirectional forging and the hot-rolling state, the intensity of the diffraction peak increases and indicates the contents of the constituent phases, but the type of phase composition remains unchanged.

Figure 3 shows the room-temperature tensile curves of the GAZ 211 alloy. The engineering stress–strain curves are shown in Figure 3a. In the as-cast state, the ultimate tensile strength (UTS), yield strength (YS), and elongation (EL) are 146 ± 4.1 MPa, 88 ± 3.4 MPa, and 14.6%, respectively. After MDF, compared with the mechanical properties in the as-cast state, the mechanical properties first increase due to the abovementioned grain refinement and intermetallic compound break-up and then decrease due to grain coarsening caused by repeated short-term heating during MDF. After the first MDF, the UTS, YS, and EL are 186 ± 6.3 MPa, 108 ± 3.5 MPa, and 13.6%, respectively. The EL of 13.6% in the first pass is slightly lower than the EL of 14.6% in the as-cast state due to the increase in UTS and YS. After the third MDF, the UTS, YS, and EL are 236 ± 8.3 MPa, 104 ± 3.5 MPa, and 22.7%, respectively. Compared with the first pass of MDF, the mechanical properties after the third MDF increase. After the sixth MDF, the UTS, YS, and EL are 218 ± 7.9 MPa, 151 ± 5.6 MPa, and 26.6%, respectively. Compared with the third pass of MDF, the mechanical properties after the sixth MDF slightly decrease due to repeated short-term heating. After the ninth MDF, the UTS, YS, and EL are 197 ± 6.1 MPa, 107 ± 3.4 MPa, and 20.3%, respectively. Compared with the sixth pass of MDF, the mechanical properties after the ninth MDF further decrease due to the increase in the number of repeated short-term heating instances. In the ninth pass, grain coarsening occurs, and strength and elongation decrease. After hot rolling (HR) following MDF, the UTS, YS, and EL are 260 ± 9.0 MPa, 171 ± 6.4 MPa, and 20.4%, respectively. The mechanical properties increase due to the rolling dislocation strengthening or strain hardening. This indicates that after MDF plus HR, the UTS of 260 ± 9.0 MPa, YS of 171 ± 6.4 MPa, and EL of 20.4% have been demonstrated in the GAZ211 alloy. The true stress–strain curves are shown in Figure 3b. With the increase in the true strain, true stress increases. Strain hardening or work hardening dominates the entire deformation process. As shown in Figure 3, the PLC effect, a serrated flow phenomenon, occurs in the stress–strain curve. The PLC effect is due to the interaction between solutes such as Gd, Al, and Zn and dislocations. It is shown that the stress drop amplitude increases, and the serration spacing is sparse with the increase in MDF passes and the reduction in the grain size. A further discussion is given in Section 4.2.

The relationship between microstructures and strength properties (load-bearing capacity) at room temperature is analyzed. As shown in Figure 1 and Figure 3, as comparison with the grain size of 53 ± 4.1 μm in the homogenized state and the UTS and YS of 146 ± 4.1 and 88 ± 3.4 MPa, respectively, in the as-cast state, the third pass of the MDF microstructure with a grain size of 26 ± 2.9 μm corresponds to the UTS and YS of 236 ± 8.3 and 104 ± 3.5 MPa, respectively. This indicates that the microstructural refinement leads to the enhancement of the strength property. After MDF plus HR, the microstructure with a grain size of 22 ± 2.5 μm corresponds to the UTS and YS of 260 ± 9.0 and 171 ± 6.4 MPa, respectively. Because of the MDF plus HR processing, microstructural refinement further raises the strength property. Meanwhile, on account of the MDF plus HR processing, dislocation density increases. What is more, the second-phase particles in the microstructures are fragmented and refined and result in the second phase strengthening due to the abovementioned processing. This is why the UTS and YS of 260 ± 9.0 and 171 ± 6.4 MPa are achieved in the rolled microstructure. In a word, the microstructural refinement results in an enhancement in the strength property.

### 3.2. Microstructure and Stress–Strain Curves of the Present Alloy at Elevated Temperature

Figure 4 presents the microstructures of the gauge section of the GAZ211 alloy at different deformation temperatures of 573–723 K and strain rates of 1.67 × 10^−2^–1.67 × 10^−4^ s^−1^. Table 1 shows the grain sizes and elongations of the GAZ211 alloy tensiled at different temperatures and strain rates. As shown in Table 1, the grain sizes are in the range of 15–42 µm, and the elongations are in the range of 108.3–228.4%, which indicates superplasticity-like behavior in coarse-grained alloy. The average elongation is 160.7%.

The microstructures in Figure 4(a1–d1) are analyzed at 1.67 × 10^−2^ s^−1^. As shown in Figure 4(a1), the microstructure is a bimodal grain size microstructure that consists of elongated coarse grains and fine equiaxed grains. The average grain size is 23 ± 4.1 µm. It is noted that DRX starts to happen. As shown in Figure 4(b1), compared with Figure 4(a1), the extent of dynamic recrystallization increases, and the number of smaller DRX grains increases. The average grain size is 15 ± 1.5 µm. It is noted that coarse grains gradually disappear, and smaller DRX grains increase. As shown in Figure 4(c1), the degree of DRX increases, but some larger grains still exist. The average grain size is 20 ± 3.0 µm. As shown in Figure 4(d1), due to the increase in temperature to 723 K, the microstructure becomes more uniform, and the degree of DRX is deepened. The average grain size is 22 ± 2.9 µm. Meanwhile, the elongation increases with the increase in the temperature at 1.67 × 10^−2^ s^−1^.

The microstructures in Figure 4(a2–d2) are analyzed at 1.67 × 10^−3^ s^−1^. As shown in Figure 4(a2), smaller grains surround the coarse grain, bimodal microstructure are still visible, and DRX occurs. The average grain size is 22 ± 2.0 µm. As shown in Figure 4(b2), the extent of DRX deepens, and the microstructure becomes more uniform. The average grain size is 19 ± 3.1 µm. As shown in Figure 4(c2), full DRX occurs. The average grain size is 17 ± 1.8 µm. As shown in Figure 4(d2), grain growth occurs. The average grain size is 35 ± 3.5 µm.

The microstructures in Figure 4(a3–d3) are analyzed at 5 × 10^−4^ s^−1^. As shown in Figure 4(a3), compared with Figure 4(a2), bimodal microstructures are still visible, DRX occurs, and the degree of DRX is deepened. The average grain size is 17 ± 2.7 µm. As shown in Figure 4(b3), the number of equiaxed grains increases, and the distribution of DRX grains becomes more uniform. The average grain size is 16 ± 2.8 µm. As shown in Figure 4(c3), full DRX and grain elongation occur. Due to the long-term heating at the lower strain rate of 5.0 × 10^−4^ s^−1^, grain growth is enhanced. The average grain size is 34 ± 3.2 µm. As shown in Figure 4(d3), grain growth further develops. The average grain size is 40 ± 4.0 µm. Further, as shown in Figure 4(c3,d3), obvious grain elongation appears, which indicates the existence of an intragranular slip of dislocations.

The microstructures in Figure 4(a4–d4) are analyzed at 1.67 × 10^−4^ s^−1^. As shown in Figure 4(a4,b4), the microstructures are still bimodal microstructures, and DRX occurs. The average grain sizes are 19 ± 3.2 and 18 ± 2.4 µm, respectively. As shown in Figure 4(c4,d4), grain elongation due to an intragranular slip of dislocations and grain growth become more obvious. The average grain sizes are 38 ± 3.4 and 42 ± 2.7 µm, respectively.

To sum up, DRX and grain growth take place during the tensile deformation at temperatures of 573–723 K and strain rates of 1.67 × 10^−2^ s^−1^–1.67 × 10^−4^ s^−1^. The black particles in Figure 4 are intermetallic compounds. They play a role in promoting DRX nucleation at high temperatures.

Figure 5 presents the engineering stress–strain curves of the GAZ211 alloy at different temperatures and initial strain rates. For a single curve, the engineering stress–strain curve exhibits a strain-hardening stage, a short-term transitional stage, and a strain-softening stage. For different curves, the engineering stress decreases with the increase in tensile temperature from 573 to 723 K and with the decrease in strain rate from 1.67 × 10^−2^ to 1.67 × 10^−4^ s^−1^. This is because with the increase in temperature and decrease in strain rate, the tensile time is prolonged, thermal activation accelerates, dislocation density decreases, and stress decreases. As shown in Figure 5, the peak stress ranges from 7 to 60 MPa as the elongation ranges from 108.3 to 228.4%. This alloy demonstrates enhanced ductility at elevated temperatures. The maximum elongation of 228.4% is demonstrated in the GAZ211 alloy at 673 K and 5.0 × 10^−4^ s^−1^.

Figure 6 shows the true stress–strain curves of the GAZ211 alloy at different temperatures and initial strain rates. Most of the flow stress curves exhibit three stages: a stress-increasing stage, a slow stress-increasing stage, and a stress-decreasing stage. At the stress-increasing stage, dislocations multiply and increase with the increase in the true strain, and strain hardening occurs. At the slow stress-increasing stage, the curves exhibit slow strain hardening with the increase in the true strain and are similar to Sotoudeh–Bate curves in the 5083 Al-Mg alloy and the Al-3.25Mg-0.37Zr-0.28Mn-0.19Y alloy named by Cao et al. [35,36]. At the stress-decreasing stage, DRX and resultant strain softening occur. Except for the case that appears in Figure 6c, the flow stress curves exhibit a stress-increasing stage and a strain-softening stage at 5.0 × 10^−4^ s^−1^ and 573 and 623 K. The slow-increasing stage almost disappears. This case indicates that the flow stress curves behave like conventional DRX curves. Overall, strain hardening, no matter whether a sharp stress increase or a slow stress increase, plays a predominant role in the flow stress curves. However, strain softening governs the deformation process at the final stage.

The relationship between microstructures and ductility at elevated temperatures is analyzed. As shown in Figure 4, Figure 5 and Figure 6, with the increase in tensile temperature and/or decrease in tensile strain rate, thermal activation accelerates, the grain size increases, and the ductility varies. This clearly indicates that tensile deformation at elevated temperatures is a thermally activated process.

### 3.3. Establishment of Constitutive Models at Elevated Temperatures

#### 3.3.1. Modified Zerilli–Armstrong Constitutive Model Incorporating the Number of Dislocations

The Zerilli–Armstrong constitutive model is based on dislocation kinetics [37]. Its building mode is similar to the Johnson–Cook constitutive model. Both equations belong to semi-empirical models. Some material parameters in the establishment of the Zerilli–Armstrong model are difficult to verify [38]. However, the modified Zerilli–Armstrong (MZA) constitutive model overcomes such disadvantages.

Here, 573 K is taken as the reference temperature, and 1.67 × 10^−4^ s^−1^ is taken as the reference strain rate. Flow stress is expressed as the following relation in the MZA model [39]:(1)$$\sigma =\left(C_{1}+C_{2}\varepsilon^{n}\right)\exp\left[-\left(C_{3}+C_{4}\varepsilon \right)T^{*}+\left(C_{5}+C_{6}T^{*}\right)\ln{\dot{\varepsilon }}^{*}\right]$$
where $T^{*}=T-T_{ref}$, ${\dot{\varepsilon }}^{*}=\dot{\varepsilon }/\dot{\varepsilon_{0}}$, n,$\;C_{1}$,$\;C_{2}$,$\;C_{3}$,$\;C_{4}$,$\;C_{5}$, $\mathrm{and}\;C_{6}\;$in Equation (1) are relevant material constants, σ is the flow stress of the GZA211 alloy during hot tensile deformation, and ε is the true strain.

When the strain rate takes the reference strain rate, ${\dot{\varepsilon }}^{*}=1\;\mathrm{is}\;\mathrm{obtained}$, and Equation (1) is simplified as:(2)$$\sigma =\left(C_{1}+C_{2}\varepsilon^{n}\right)\exp\left[-\left(C_{3}+C_{4}\varepsilon \right)T^{*}\right]$$

A logarithm is taken on both sides of Equation (2), and Equation (2) becomes:(3)$$\ln\sigma =\ln\left(C_{1}+C_{2}\varepsilon^{n}\right)-\left(C_{3}+C_{4}\varepsilon \right)T^{*}$$

The true strain and true stress of the GAZ211 alloy at 1.67 × 10^−4^ s^−1^ are substituted into Equation (3), and linear fitting is performed. The relation of lnσ-$T^{*}\;$is obtained. Furthermore, the slope and intercept of the corresponding fitting lines are obtained, where the value of $\ln\left(C_{1}+C_{2}\varepsilon^{n}\right)\;$is the linear intercept, and the value of $-\left(C_{3}+C_{4}\varepsilon \right)\;$is the linear slope.

Figure 7 presents the linear fitting relation between lnσ and $T^{*}$. The lnσ values of four fitting lines decrease with the increase in $\;T^{*}$. When $T^{*}\;$are identical, the lnσ values corresponding to different strain rates differ little.

A new parameter,$\;\;J_{1}$, is introduced to Equation (3), and the linear intercept is expressed as:(4)$$J_{1}=\ln\left(C_{1}+C_{2}\varepsilon^{n}\right)$$

When a logarithm is taken on both sides of the converted Equation (4), the following is obtained:(5)$$\ln\left(\exp J_{1}-C_{1}\right)=\ln C_{2}+n\ln\varepsilon$$
where $C_{1}\;$can be substituted by the yield stress under the reference deformation condition; here,$\;C_{1}=38.00$ MPa. Corresponding strain rates and the linear intercept obtained by fitting in Figure 7 are substituted into Equation (5), and the relation curve of $\ln\left(\exp J_{1}-C_{1}\right)\;$and lnε can be obtained, where the linear slope is the *n*-value, and the intercept is the $\ln C_{2\;}$ value. Conversion to$\;\ln C_{2}$ is performed, and the $\;C_{2}\;$value is obtained.

Figure 8 is the linear fitting curve between $\ln\left(\exp J_{1}-C_{1}\right)\;$ and lnε based on Equation (5) and relevant data. The correlation coefficient *R* = 0.99649. The linear slope n =1.71943, and $C_{2}=3.12912$ MPa.

To solve constants $\;C_{3}\;$ and $\;C_{4}$, a new parameter, $\;S_{1}$, is introduced to make it equal to the slope of Equation (3). Then, the following is obtained:(6)$$S_{1}=-\left(C_{3}+C_{4}\varepsilon \right)$$

The slopes and relevant strains in various linear fitting lines as per Figure 7 are substituted into Equation (6), respectively, and the scatter diagram between $\;S_{1}$ and ε is obtained. Then, the scatter diagram is fitted linearly, and one obtains the linear slope $\;C_{4}$ and the linear intercept$\;C_{3}$.

Figure 9 presents the linear relation between $S_{1}$ and ε. The correlation coefficient *R* = 0.99931, indicative of an excellent fitting effect. Moreover, $C_{3}\;$and $C_{4}$ can be obtained according to the slope and intercept. $C_{3}=0.01054$, and $\;C_{4}=-0.0027$.

To solve material constants $\;C_{5}\;$ and $\;C_{6}$, the logarithm is taken on both sides of Equation (1).
(7)$$\ln\sigma =\ln\left(C_{1}+C_{2}\varepsilon^{n}\right)-\left(C_{3}+C_{4}\varepsilon \right)T^{*}+\left(C_{5}+C_{6}T^{*}\right)\ln{\dot{\varepsilon }}^{*}$$

The true strain, flow stress, temperature, and solved material constants, n,$\;C_{1}$,$\;C_{2}$,$\;C_{3}$,$\;\mathrm{and}\;C_{4}$, are substituted into Equation (7)*,* and the linear slopes and intercepts of the curves between lnσ and$\;\ln{\dot{\varepsilon }}^{*\;}$are obtained. In this case, the slope is the value of $C_{5}+C_{6}T^{*}$.

Then, another new parameter, $\;S_{2}$, is introduced to make it equal to the slope of Equation (7). Hence, the$\;S_{2}$ value is expressed as:(8)$$S_{2}=C_{5}+C_{6}T^{*}$$

Figure 10 presents the linear relation between lnσ and$\;\ln{\dot{\varepsilon }}^{*}$ based on Equation (7). The correlation coefficient *R* and the linear slope increase with the increase in the temperature. The range of *R* and the linear slope are 0.953719~0.99591 and 0.13392~0.17590, respectively.

The slope in Figure 10 is substituted into Equation (8), and the scatter distribution relation between$\;S_{2}\;$and $\;T^{*}$ is obtained. Then, the scatter points are fitted linearly, and the linear relation is shown in Figure 11. The slope and intercept in Figure 11 are $C_{6}$ and $C_{5}$, respectively. Hence, $C_{5}=0.15604$, and $C_{6}=0.00013$.

Table 2 shows the parameters for the MZA model. The relevant parameters are substituted into Equation (1), and the MZA model for the GAZ211 alloy is obtained as the following:(9)$$\sigma_{MZA}=\left(38.00+3.13\varepsilon^{1.71943}\right)e^{\left[-\left(0.01054-0.0027\varepsilon \right)T^{*}+0.00013\ln{\dot{\varepsilon }}^{*}\right]}=F\left(Z\right)\;$$

To validate Equation (9), true stress versus true strain curves with measured values and calculated values are plotted, as shown in Figure 12.

In most cases, the error between the calculated stress and measured stress decreases with an increase in the temperature and/or a decrease in the strain rate. An abnormal case occurs at 623 K, which remains to be investigated in the future. The average absolute relative error is 9.63% with a good prediction. In particular, the MZA model has good predictability at the reference strain rate.

Based on our previous work on the model of the number of dislocations [40], substituting Equation (9) into this model leads to the following dislocation model incorporating the MZA model:*N* = 1.81(1 − *ν*)[*d*/(*Gb*)] *F*(*Z*)(10)
where *N* is the number of dislocations inside a grain, *ν* is Poisson’s ratio, *d* is the linear intercept grain size, *G* is the shear modulus, *b* is the magnitude of the Burgers vector, and *F*(*Z*) is the flow stress obtained by Equation (9). Since the model of the number of dislocations established is not estimated in reference [40], an estimation is made to disclose the underlying mechanism in the present GAZ211 alloy. Thus, according to Equation. (10), the number of dislocations can be calculated at various temperatures and strain rates.

*G* is given by the following relation [41]:*G* = *E*/[2(1 + *ν*)]
*E* = 48,700 − 8.59*T* − 0.0195*T*^2^(11)

An estimation was made for the case in which an elongation of 228.4% is obtained. As per Equation (11), *G* = 12,624.75 MPa. With the substitution of *T* = 673 K, $\dot{\varepsilon }$ = 5 × 10^−4^ s^−1^, *F*(*Z*) = 16 MPa (Equation (9)), *d* = 3.4 × 10^−5^ m (Figure 4(c3)), *ν =* 0.35 [42], and *b* = 3.21 × 10^−10^ m [43] into Equation (10), one obtains *N* = 145.78 ≈ 146. This means that there are 146 pieces of dislocations inside a grain when an elongation of 228.4% is obtained. The calculation is indicative of significant dislocation activity in this alloy under this condition.

#### 3.3.2. Power-Law Constitutive Model

The high-temperature power-law constitutive equation of the GAZ211 alloy will be established through determination of the stress exponent, the grain size exponent, and the deformation activation energy.

The high-temperature power-law constitutive equation is generally given by [44,45]:(12)$$\dot{\varepsilon }=\frac{AD_{0}Gb}{kT}{\left(\frac{b}{d}\right)}^{p}{\left(\frac{\sigma -\sigma_{0}}{G}\right)}^{n}\exp\left(-\frac{Q}{RT}\right)$$
where $\dot{\varepsilon }$ is the steady-state deformation rate, *A* is a dimensionless constant, *G* is the shear modulus, *b* is the magnitude of the Burgers vector, *p* is the grain size exponent, *σ* is the applied stress, *n* is the stress exponent (1/*m*, *m*-strain rate sensitivity index or *m*-value), *D*_0_ is the frequency factor for diffusion, *Q* is the deformation activation energy, *k* is Boltzmann’s constant, *T* is the absolute temperature, *d* is the grain size, and *R* is the universal gas constant.

(1) Stress exponent *n*

The stress exponent *n* is related to the threshold stress $\sigma_{\mathrm{th}}$. The threshold stress caused by the second phase particle is the onset stress to initiate the plastic flow. At the true strain of 0.2, the values of threshold stress $\sigma_{0}$ are determined at zero strain rate using linear fitting of the σ −$\;\dot{\varepsilon }$^1/*n*^ relation. Threshold stress is not a constant value at all temperatures. It decreases with the increase in temperature. True stress at a true strain of 0.2 is chosen from the flow stress curves. Then, $\sigma -{\dot{\varepsilon }}^{1/2},\sigma -{\dot{\varepsilon }}^{1/3},\sigma -{\dot{\varepsilon }}^{1/4},\;\sigma -{\dot{\varepsilon }}^{1/5},\;\sigma -{\dot{\varepsilon }}^{1/6},\;$and$\;\sigma -{\dot{\varepsilon }}^{1/7}$ curves are fitted linearly. The intercepts of the linear curves are the threshold stresses, respectively, when *n* is 2, 3, 4, 5, 6, and 7. When the $\sigma_{\mathrm{th}}$ value is positive, and the fitting quality or correlation coefficient is the best, the corresponding *n* value is the optimal value.

Figure 13 shows the fitting curves of $\sigma -{\dot{\varepsilon }}^{1/2},\sigma -{\dot{\varepsilon }}^{1/3},\sigma -{\dot{\varepsilon }}^{1/4},\;\sigma -{\dot{\varepsilon }}^{1/5},\;\sigma -{\dot{\varepsilon }}^{1/6},\;$and$\;\sigma -{\dot{\varepsilon }}^{1/7}$, respectively. The threshold stress is negative when the *n* values are 5, 6, and 7. Thus, *n* = 5, 6, and 7 are excluded. The correlation coefficients are 0.9568, 0.9738, and 0.9812, respectively, when the *n* values are 2, 3, and 4. It is found that the average *R* value is the largest when *n* = 4. Hence, *n* = 4 is the best stress exponent or the true stress exponent.

(2) Deformation activation energy *Q*

The true deformation activation energy at a constant strain rate is expressed as the following relation as per Equation (12):(13)$$Q=R{\frac{\partial \left[\ln\left(\sigma^{n}G^{1-n}T^{-1}d^{-p}\right)\right]}{\partial \left(T^{-1}\right)}|}_{\dot{\varepsilon }}$$

As the elongations of the present alloy are 108.3–228.4% and exhibit quasi-superplasticity [46], *p* = 0.

Figure 14 presents the fitting curves of $\ln\left(\sigma^{n}G^{1-n}T^{-1}\right)-1/T$ at different strain rates. The deformation activation energy for the GAZ211 alloy is in the range of 90.33–121.17 kJ/mol. The average activation energy is 104.42 kJ/mol. The changing feature of activation energy will be discussed in Section 4.5.

(3) Normalized curve

Equation (12) is turned into the following form by taking the logarithm:(14)$$\ln\left(\frac{\dot{\varepsilon }kT}{DGb}\right)=\mathrm{lnA}+n\ln\frac{\sigma -\sigma_{0}}{G}$$
where $D=D_{0}\exp\left[-Q/RT\right]$, *D*_0_ = 1 × 10^−4^ m^2^ s^−1^ [47], *Q* is the average deformation activation energy, and Q = 104.42 kJ/mol. Hence,$\;D=10^{-4}\exp\left[-104,420/RT\right]$.

Figure 15 presents the fitting curve of $\ln\left(\dot{\varepsilon }kT/DGb\right)-\ln\left[\left(\sigma -\sigma_{0}\right)/G\right]$. The slope of the fitting line is 4. The intercept of the fitting line is lnA (=39.25). Hence, A = 1.11 × 10^17^. The correlation coefficient *R* is 0.9695.

Thus, the power-law constitutive equation of the GAZ211 alloy is obtained:(15)$$\dot{\varepsilon }=1.11\times 10^{17}\frac{Gb}{KT}{\left(\frac{\sigma -\sigma_{0}}{G}\right)}^{4}D$$

Finally, the relationship between the average grain size and the Zener–Hollomon parameter (*Z* parameter) needs to be established. The *Z* parameter is given by *Z* =$\;\dot{\varepsilon }\;\exp\left[Q/\left(RT\right)\right]$, where the meaning of symbols is the same as in Equation (12).

It is presumed that $d=aZ^{n^{\prime }}$, where a and *n*′ are constants. Taking the logarithm on both sides, the following is obtained:(16)$$lnd=\mathrm{lna}+n^{\prime }\ln Z$$

Figure 16 presents the plot of ln*d* against ln*Z*, where a = 12.3424, and *n*′ = −0.0688. Thus, the *d-Z* model is obtained:(17)$$d=12.3424Z^{-0.0688}$$

As shown in abovementioned results, errors are reflected in the measurement of grain size, the measurement of mechanical properties, and modeling. The errors include the instrument measurement error, the human naked eye examination error, and the calculation rounding-off error. During the measurement of grain size and mechanical properties, errors inevitably occur. Thus, normal maintenance of instruments such as the optical microscope and tensile tester is necessary, and scientific training for the operating researcher is also necessary to keep the errors within a permissible scope. These errors indirectly reflect the experimental level that our institution attains.

## 4. Discussion

### 4.1. Estimation of Room-Temperature Strengthening and Contribution to Yield Strength Property in GAZ211 Alloy

The GAZ211 alloy undergoes different processing stages, such as casting, homogenization, MDF, and hot rolling. During the alloying process, solutes such as Gd, Al, and Zn existing in the Mg matrix cause lattice strain or distortion, increase the resistance of dislocation movement in the lattice, and lead to solid solution strengthening. During the MDF and hot-rolling process, since grains and second-phase particles are fragmented and refined, Hall–Petch grain refinement strengthening and second-phase strengthening occur. In the meantime, MDF and hot rolling restrict the movement of dislocations, increase the dislocation density and the deformation resistance, and lead to strain hardening or dislocation strengthening. Therefore, it is essential to estimate the strengthening stress of various strengthening mechanisms, calculate the contribution of individual mechanisms to the yield strength property, and deepen the understanding of the room-temperature strengthening mechanism.

The alloy strengthening yield strength$\;\sigma_{y}$ includes intrinsic stress$\;\sigma_{0}$, dislocation strengthening stress$\;\sigma_{d}$, grain boundary strengthening stress$\;\sigma_{gb}$, solid solution strengthening stress $\sigma_{ss}$, and second-phase strengthening stress$\;\sigma_{Oro}$, as shown in Equation (18):(18)$$\sigma_{y}=\sigma_{0}+\sigma_{d}+\sigma_{gb}+\sigma_{ss}+\sigma_{Oro}$$

As shown in Figure 3a, the experimental yield strength of the GAZ211 alloy is 171 MPa.

(1) Contribution of intrinsic stress to the yield strength

The intrinsic stress is the Peierls–Nabarro stress to impede dislocation movement on the slip plane. The intrinsic stress $\sigma_{0}=11\;\mathrm{MPa}$ according to the stress range of 8–14 MPa [48]. Hence, the contribution of intrinsic stress to the yield strength is 6.43%.

(2) Contribution of dislocation strengthening to the yield strength

MDF and hot-rolling deformation induce a large number of dislocations, increase dislocation density, and lead to dislocation strengthening. The dislocation strengthening stress is given by [49]:(19)$$\sigma_{d}=\alpha_{d}Gb\sqrt{\rho }$$
where$\;\alpha_{d}$ is a constant, 1.04 [49], *b* is the magnitude of the Burges vector, ρ is the dislocation density of the hot-rolled alloy, and *G* is the shear modulus, 16,661.62 MPa at room temperature of 293 K.

The dislocation density ρ is given by [50]:(20)$$\rho =\frac{2\sqrt{3}e^{2}{}^{1/2}}{bd^{\prime }}$$
where $d^{\prime }$ is the coherent diffraction domain size, and e is the lattice strain.

Based on our previous method to determine the dislocation density in a hot-rolled plate of the Mg-3Li-3Al-3Zn-0.5Y alloy using XRD peak broadening [51] and Figure 2 in Section 3.1, we obtained$\;\rho =5.72\times 10^{13}$ m^−2^ in this alloy. Substitution of the abovementioned data into Equation (19) gives $\sigma_{d}=$41.91 MPa. Hence, the contribution of dislocation strengthening to the yield strength is 24.51%.

(3) Contribution of grain boundary strengthening to the yield strength

MDF and hot-rolling deformation result in grain refinement. The grain refinement increases the number of grain boundaries, leads to the pile-up and stress concentration of dislocations at the head of the grain boundary, and increases the resistance to dislocation motion. Thus, grain boundary strengthening or Hall–Petch strengthening occurs. The grain boundary strengthening stress is given by [49]:(21)$$\sigma_{gb}=\alpha_{HP}G\sqrt{b/d}\;$$
where $\alpha_{HP}$ is a constant, 0.413 [49], and *d* is the grain size, 2.2 × 10^−5^ m (Figure 1c). Hence, $\sigma_{gb}$ = 26.29 MPa. Thus, the contribution of grain boundary strengthening to the yield strength is 15.37%.

(4) Contribution of solid solution strengthening to the yield strength

In the present alloy, solid solution strengthening results from the interaction between dislocations and solutes of Gd, Al, and Zn elements inside the grain. This is because lattice misfit strain and associated distortion stress in the grain interior caused by the processing history hinder the dislocation motion and lead to the increment in yield strength, i.e., the strength property. $\sigma_{ss}$ is given by [52,53]:(22)$$\sigma_{ss}=MGb\varepsilon_{ss}^{2/3}c^{1/2}$$
where *M* is the Taylor factor of texture-containing pure Mg, 2.5 [54], *G* is the shear modulus, *b* is the magnitude of the Burges vector, $\varepsilon_{ss}\;$is the lattice distortion strain, =[*r*_s_ − *r*_Mg_]/*r*_Mg_, where *r*_s_ is the atomic radius of the solute, *r*_Mg_ is the atomic radius of the solvent, and *c* is the concentration of the element. Substitution of the abovementioned parameters into Equation (22) gives $\sigma_{ss}$ = 37.99 MPa. Thus, the contribution of solid solution strengthening to the yield strength is 22.22%.

(5) Contribution of Orowan strengthening to the yield strength

After MDF and hot rolling deformation, the second-phase particles are crushed into smaller particles, hindering the dislocation motion and raising the strength. The Orowan bypassing mechanism, instead of the cutting mechanism, is often claimed in a majority of Mg-Gd alloys [55]. Hence, Orowan strengthening stress is obtained as per Equation (18):(23)$$\sigma_{Oro}=\sigma_{y}-\left(\sigma_{0}+\sigma_{d}+\sigma_{gb}+\sigma_{ss}\right)$$

The abovementioned data are substituted into Equation (23), and we obtain $\sigma_{Oro}$ = 53.82 MPa. Therefore, the contribution of Orowan strengthening to the yield strength is 31.47%. This indicates that Orowan strengthening makes a significant contribution to the yield strength.

In terms of the aforementioned estimations, it is noted that the second-phase strengthening and dislocation strengthening contribute to a large fraction of the yield strength, whereas grain boundary strengthening contributes to a smaller fraction of the yield strength. The cause of the larger contribution of the second-phase strengthening and dislocation strengthening is that the second-phase particles such as (Mg, Al)_3_Gd, Al_7_Zn_3_, and Al_2_Gd intermetallic compounds in Figure 2 achieve effective fragmentation or break-up, and MDF and hot rolling increase dislocation density and result in pronounced strain hardening. The cause of higher Orowan stress is that the deformation-crushed particles of Al_2_Gd and (Mg, Al)_3_Gd with high melting points of 1798 K and 823 K, respectively, act as efficient hard particles to raise the resistance of dislocation motion and enhance the yield strength. In particular, the Al_2_Gd particle is similar to the Al_2_Y particle with a high melting point of 1758 K in the Mg-Gd-Y-Al alloy and plays an important role in the enhancement of the strength property. The cause of the smaller contribution of grain boundary strengthening to the yield strength property is that the coarse grain size (grain size >10 μm) following MDF reduces the efficacy of grain boundary strengthening or decreases grain boundary strengthening. This issue remains to be examined during MDF deformation in the future.

It is worth mentioning that load transfer stress and thermal mismatch stress are often estimated in metal matrix composites, and the present GAZ211 alloy is a multicomponent alloy instead of a composite. Hence, the load transfer stress and thermal mismatch stress are not considered in this alloy. The calculation here only gives a rough estimation of the fractions of different contributions. For more quantitative analysis, better microstructural characterization is required, e.g., for Orowan strengthening in this alloy, one should determine the relevant parameters of the particle size and volume fraction of particles instead of using the subtraction calculation.

### 4.2. Analysis of Type C PLC Effect at Room Temperature

The PLC effect, a jerky flow in the stress–strain curves, includes five types or modes: A, B, C, D, and E [56]. Different PLC effects possess different PLC effect propagation methods. According to the relevant literature [57,58], Type A belongs to locking serration and exhibits a sudden rise followed by a drop. Type B belongs to oscillating serration and exhibits intermittent propagation with regular oscillations. Type C belongs to unlocking serration and exhibits an irregular sudden stress drop below the general stress level with a serration amplitude larger than type A and type B. Type D and type E are similar to type A but do not exhibit strain hardening or work hardening. In terms of the aforementioned types of serration, as shown in Figure 3 in Section 3.1, the PLC effect in GAZ211 processed by MDF belongs to type C. The type C serration is randomly distributed in the stress–strain curves and does not have a fixed spatial correlation. Once Cottrell’s lockings between the solutes such as Gd, Al, and Zn and the dislocations are released, and unlocking starts to appear, the stress suddenly drops. After the stress drops to the lowest point, a stress drop amplitude forms. With the increase in tensile strain, dislocations multiply, new strain hardening occurs, and the drop stress starts to rise until the general stress level. Hence, a “V”-shaped serration appears. This process repeats, which leads to the appearance of sparse type-C serrations. This is the thought behind type-C serration or the PLC effect in the GAZ211 alloy processed by MDF.

The PLC effect or serrated flow is associated with the fragmentation and refinement of grains and second-phase particles, as well as the variation in dislocation density during MDF. As shown in Figure 3 in Section 3.1, with the increase in MDF passes, grains and second phases are refined, and the dislocation density increases. As a result, the interaction between solutes and dislocations is enhanced, and the interaction between second-phase particles and dislocations is also increased. The pinning and depinning become stronger, and the stress drop amplitude increases with the increase in MDF passes. This results in a higher stress drop amplitude with the progress of MDF. Once the locking state is changed into the unlocking state, the abovementioned “V-shaped” serration or type C serration occurs. In addition, the serration spacing is sparse, which is because MDF grain refinement reduces the opportunity for pinning and depinning. This evidence is probably the influential cause for type-C serrations being observed in the stress–strain curves.

### 4.3. Analysis of Microstructural and Flow Stress Characteristics at Elevated Temperature

The microstructural characteristics were analyzed. As shown in Figure 4 in Section 3.2, first, some microstructures are bimodal microstructures consisting of fine grains and coarse grains at a strain of 1.67 × 10^−2^ s^−1^. The bimodal microstructures belong to incomplete DRX microstructures. Usually, necessary rolling reduction is required to induce the occurrence of complete DRX, e.g., a reduction of more than 70%. The big reduction provides necessary storage deformation energy to promote DRX nucleation. However, in the GAZ211 alloy, to avoid the occurrence of edge cracks of rolled plate, the reduction is 34% and not sufficient to induce complete DRX, which is consistent with the report on incomplete recrystallization in AZ31 magnesium alloy [59,60]. This is the cause of the formation of the incomplete DRX microstructure. Moreover, a small number of fine grains (grain size ≤ 10 μm) undergo grain boundary sliding, whereas a majority of coarse grains (grain size > 10 μm) undergo intragranular slip or climb during tensile deformation at elevated temperatures. Second, complete DRX is achieved at strain rates of 1.67 × 10^−3^–1.67 × 10^−4^ s^−1^at higher temperatures. This is because thermal activation exceeds the hardening, and coarse grains are refined by continuous DRX with the transformation of the low-angle grain boundary in coarse grains into a high-angle grain boundary during tension at higher temperatures. In this case, the microstructure of complete or full DRX is thermally stable. Third, some dynamic grain growth phenomena occur after complete DRX at temperatures of 673–723 K at slower strain rates. Deformation-induced grain growth and thermal static grain growth occur simultaneously under these conditions. In this case, the microstructure of dynamic grain growth is thermally unstable. In terms of the aforementioned analysis, the microstructural characteristics for tensile deformation at elevated temperatures are the occurrence of incomplete DRX, complete DRX, and dynamic grain growth. In addition, the cavity morphology was observed. At higher strain rates, there is no cavity throughout the gauge length. At lower strain rates, there are only a few cavities near the fracture tip.

The flow stress characteristic was analyzed. As shown in Figure 6 in Section 3.2, the flow stress curves exhibit three stages: a sharp stress-rising stage, a long and slow stress-rising stage, and a short stress-decreasing stage. The sharp stress-rising stage corresponds to elastic and yield deformation with the generation and multiplication of dislocations. The long and slow stress-rising stage corresponds to strain-hardening deformation. Although DRX occurs in this stage, thermal softening occurs, but since the strain hardening still surpasses strain softening, strain hardening still predominates. The short stress-decreasing stage corresponds to the strain-softening stage. Long-term thermal activation leads to the strain softening, although DRX followed by dynamic grain growth takes place. The characteristic of the flow stress curves is that the present GAZ211 magnesium alloy behaves like an Al-Mg aluminum alloy [35,36] and exhibits Sotoudeh–Bate curves. The possible cause is that the interaction between solutes such as Gd, Al, and Zn and dislocations increase the resistance to deformation and promote the formation of Sotoudeh–Bate curves in the present GAZ211 alloy.

### 4.4. Comparison of MZA and Power-Law Constitutive Models Established in This Alloy

As shown in Section 3.3, the MZA and power-law constitutive models established in this alloy have several characteristics. First, on the one hand, the MZA constitutive model considering the number of dislocations changes the former MZA phenomenological equation into a semi-physics-based constitutive equation. This makes it possible to reflect the relationship between temperature and strain rate as well as link dislocation quantification to the temperature and strain rate for the first time in this GAZ211 alloy. On the other hand, apart from the reflection of the relationship between stress, temperature, and strain rate, the power-law constitutive model also has the nature of a physics-based constitutive equation or a microstructural-based constitutive equation due to the incorporation of relevant physical constants and grain size, a microstructural parameter. Second, in consideration of the initial coarse grain size, 22 ± 2.5 μm, the MZA constitutive model is suitable considering the number of dislocations for application to the hot forming process with typical strain rates of 10^−4^~10 s^−1^, whereas the power-law constitutive model is suitable for superplasticity and dislocation creep with typical strain rates of 10^−4^~10^−2^ s^−1^. Third, the MZA constitutive model has an average absolute relative error (AARE) of 9.63% and correlation coefficients of *R* higher than 0.9 during parameter determination, and the power-law constitutive model has a correlation coefficient of *R* (=0.9695). The statistic quantity of AARE and *R* has good predictability, which is indicative of the rationality of established models. Fourth, the model of the number of dislocations is convincing, because this model has been validated experimentally in our previous report on Al-Mg-Er-Zr alloy during hot compression [61].

Furthermore, a relevant constitutive discussion is as follows. Firstly, according to our survey of literature on constitutive modeling at elevated temperatures, a majority of reports are available on establishing the Arrhenius hyperbolic sine law, and some reports are available on the establishment of the modified Johnson–Cook equation, the modified Zerilli–Armstrong equation, the Fields–Beckofen constitutive equation, and so forth. The major reason for this is the high prediction ability of the Arrhenius hyperbolic sine law. Secondly, Mirzadeh’s and Mahamudi’s research groups [62,63] have tried to introduce the theoretical activation energy and stress exponent from creep theory to the Arrhenius hyperbolic sine law in Mg-Gd-(Zn) alloys. Their reports make a contribution to the established connection between creep characteristic values and the Arrhenius hyperbolic sine law. Thirdly, Lin et al. [64] and Kim et al. [65] have reported on the dislocation density-based constitutive equation in Ti55511 alloy and link creep equations in Prasad’s processing map, respectively. In addition, a few constitutive equations with new ideas have occasionally been proposed, and some reports on artificial intelligence such as ANN models have appeared. To a certain extent, the aforementioned results foreshow the developing trend of the study of constitutive equations. As for our work on the MZA model considering dislocation quantification, the present work is a first attempt to establish a connection between the deformation condition such as temperature and strain rate and dislocation quantification. The striking advantage of dislocation quantification is to save on the expensive experimental cost of numerous transmission electron electroscope examinations. Grain size and dislocation are two key variables during high-temperature deformation. *d*-*Z* modeling reflects the relationship between grain size and the Zener parameter. The modeling of the number of dislocations and the MZA, together with the grain size model, solves two key variables. This is expected to be implemented into practical engineering processes in the future.

### 4.5. Deformation Mechanism of the Present Alloy at Elevated Temperature

As shown in Section 3.3.2, the stress exponent *n* = 4, and the deformation activation energy *Q* = 104.42 kJ/mol. The theoretical activation energy of grain boundary diffusion is 92 kJ/mol, and the theoretical activation energy of lattice diffusion is 135 kJ/mol [43]. The experimental activation energy is slightly higher than the theoretical activation energy of the grain boundary diffusion. Meanwhile, the pipe diffusion coefficient is equal to the grain boundary diffusion coefficient, $D_{p}\;$=$\;D_{gb}$ [66]. Thus, the diffusion mechanism is pipe diffusion. To check whether grain boundary sliding controlled by pipe diffusion is the rate-controlling mechanism, we perform the following estimation to see if the Ruano et al. model [66] is consistent with the experiment strain rate:(24)$$\dot{\varepsilon }=3.2\times 10^{11}\alpha \left(D_{p}/d^{2}\right){\left(\sigma /E\right)}^{4}$$where$\dot{\;\varepsilon }$ is the strain rate, α is a material constant, $D_{p}$ is the pipe diffusion coefficient, *d* is the grain size, σ is the applied stress, and *E* is Young’s modulus. *T* = 673 K, α = 4, $D_{p}\;$=$\;D_{gb}$ = 5.633 × 10^−10^ m^2^s^−1^ calculated by reference [43], *d* = 3.4 × 10^−5^ m (Figure 4(c3)), σ = 12 MPa (Figure 6c)*,* and *E* = 34,086.81 MPa calculated by reference [41]. Substitution of the aforementioned data into Equation (24) yields $\dot{\;\varepsilon }$ = 3.034 × 10^−3^ s^−1^. The experimental strain rate at an elongation of 228.4% is 1.52 × 10^−4^ s^−1^, which is calculated by the formula ($\;\dot{\varepsilon }={\dot{\varepsilon }}_{0}/\exp\left(\varepsilon \right)$), where ${\dot{\varepsilon }}_{0}$ is the initial strain rate, 5.0 × 10^−4^ s^−1^, and ε is the true strain. Hence, the theoretical strain rate is one order of magnitude higher than the experimental strain rate. Thus, grain boundary sliding controlled by pipe diffusion is ruled out or excluded. Considering the number of dislocations inside a grain (146 pieces), *n* = 4, and *Q* = 104.42 kJ/mol, we can conclude that the rate-controlling mechanism under this condition is dislocation motion controlled by pipe diffusion.

Finally, the changing feature of activation energy in Figure 14 is discussed. At 1.67 × 10^−2^ and 1.67 × 10^−3^ s^−1^, the experimental activation energy is 95.04 and 90.33 kJ/mol, respectively, which is adjacent to the Mg pipe diffusion activation energy of 92 kJ/mol. This indicates that the diffusion is controlled by pipe diffusion. At 5.0 × 10^−4^ s^−1^, the experimental activation energy is 111.13 kJ/mol, which is higher than the Mg pipe diffusion activation energy of 92 kJ/mol but lower than the Mg self-diffusion activation energy of 135 kJ/mol. This indicates that in this case, the diffusion mechanism is pipe diffusion. At 1.67 × 10^−4^ s^−1^, the experimental activation energy is 121.17 kJ/mol, which is close to the Mg self-diffusion activation energy of 135 kJ/mol, indicating that in this case, the diffusion mechanism is self-diffusion or lattice diffusion. The aforementioned analysis shows that pipe diffusion dominates the diffusion process at elevated temperatures. During the high-temperature tensile process, atomic diffusion, as a thermal activation process, helps soften possible hardening, promotes dislocation motion, and facilitates strain or ductility enhancement.

## 5. Conclusions

(1) A lightweight novel Mg-1.85Gd-0.64Al-0.62Zn alloy was fabricated by innovative multidirectional forging and hot rolling. An ultimate tensile strength of 260 MPa, yield strength of 171 MPa, and elongation of 20.4% were demonstrated in the present alloy. Room-temperature strengthening mechanisms were identified by strengthening the model estimation. The type C Portevin-Le Chatelier effect was discovered at room temperature and elucidated in this alloy.

(2) Examination of the microstructure at elevated temperatures revealed that dynamic recrystallization and dynamic grain growth occur. In particular, it was discovered that bimodal microstructures or incomplete dynamic recrystallization microstructures exist in the high-temperature deformation.

(3) A maximum elongation of 228.4% was demonstrated in the GAZ211 alloy at 673 K and 5.0 × 10^−4^ s^−1^. Flow stress curves show that the present alloy exhibits Sotoudeh–Bate curves or a long intermediate strain-hardening stage, followed by a final strain-softening stage.

(4) The modified Zerilli–Armstrong constitutive equation incorporating the number of dislocations was established. The power-law constitutive equation was established to identify the deformation mechanism. At 673 K and 5.0 × 10^−4^ s^−1^, the stress exponent was 4, and the average deformation activation energy was 104.42 kJ/mol. The number of dislocations inside a grain was 146. This evidence confirms that dislocation motion controlled by pipe diffusion dominates the rate-controlling process under this condition.

This work has presented the first report on the microstructure, mechanical properties, modeling, and mechanism of Mg-1.85Gd-0.64Al-0.62Zn alloy at room and elevated temperatures fabricated by our innovative multidirectional forging and hot-rolling (MDF + HR) technique. The limitations to emerging multicomponent Mg-Gd system alloys lie in deformability, which prevents the occurrence of edge cracks of rolled plates, and the research and development of a viable MDF and HR schedule for emerging multicomponent Mg-Gd alloys will provide a prerequisite for industrial applications. The strengthening models and constitutive models will be put into effect in the engineering control system.

## Figures and Tables

**Figure 1 materials-16-01639-f001:**
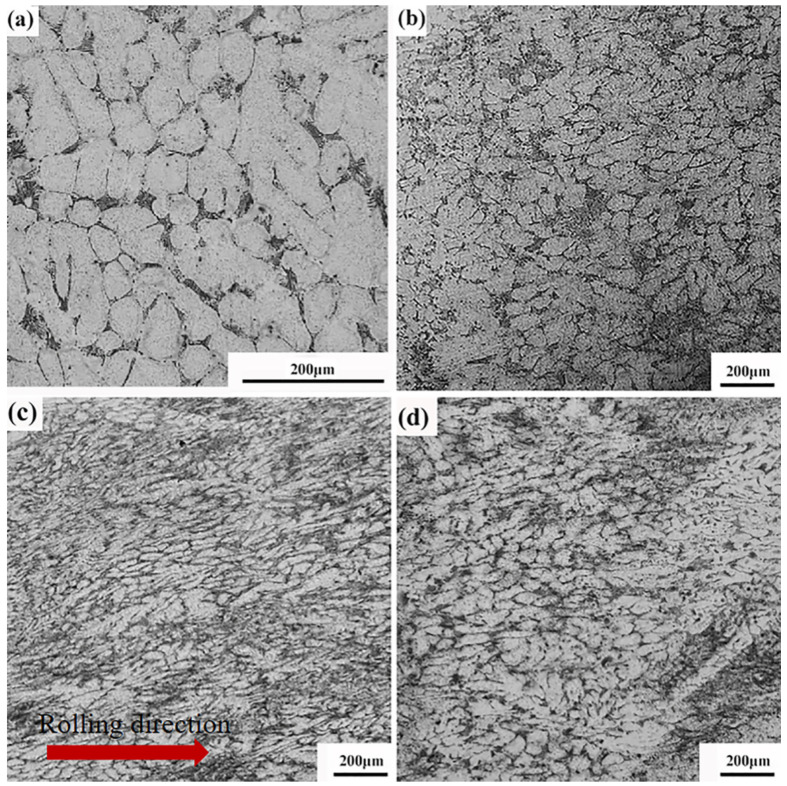
Microstructures of GAZ 211 alloy in different states: (**a**) homogenization at 673 K for 16 h, (**b**) multidirectional forging (723 K), 3rd pass, (**c**) multidirectional forging (723 K) and hot rolling (693 K), longitudinal direction (rolling direction), and (**d**) annealing at 723 K for 1 h.

**Figure 2 materials-16-01639-f002:**
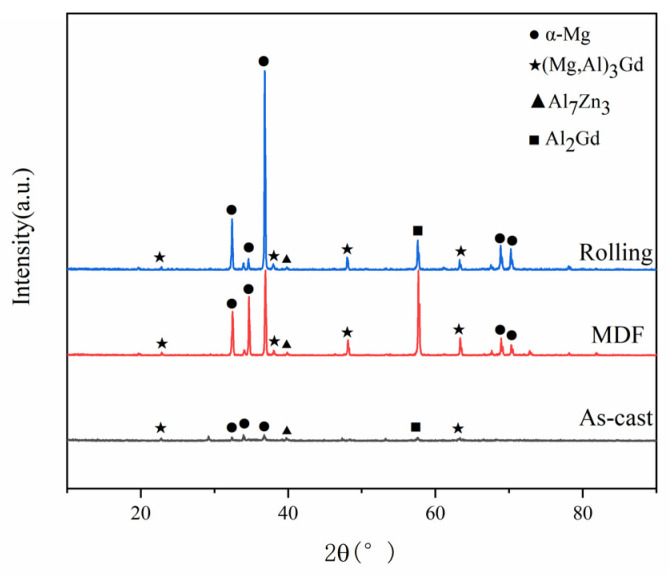
X-ray diffraction patterns of GAZ 211 alloy under different states.

**Figure 3 materials-16-01639-f003:**
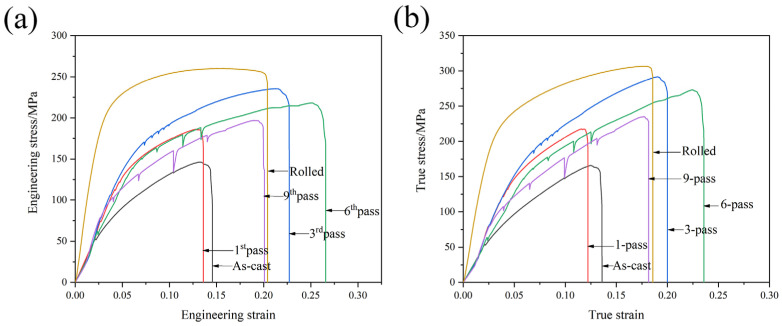
Room temperature tensile curves of GAZ 211 alloy: (**a**) engineering stress–strain curves and (**b**) true stress–strain curves.

**Figure 4 materials-16-01639-f004:**
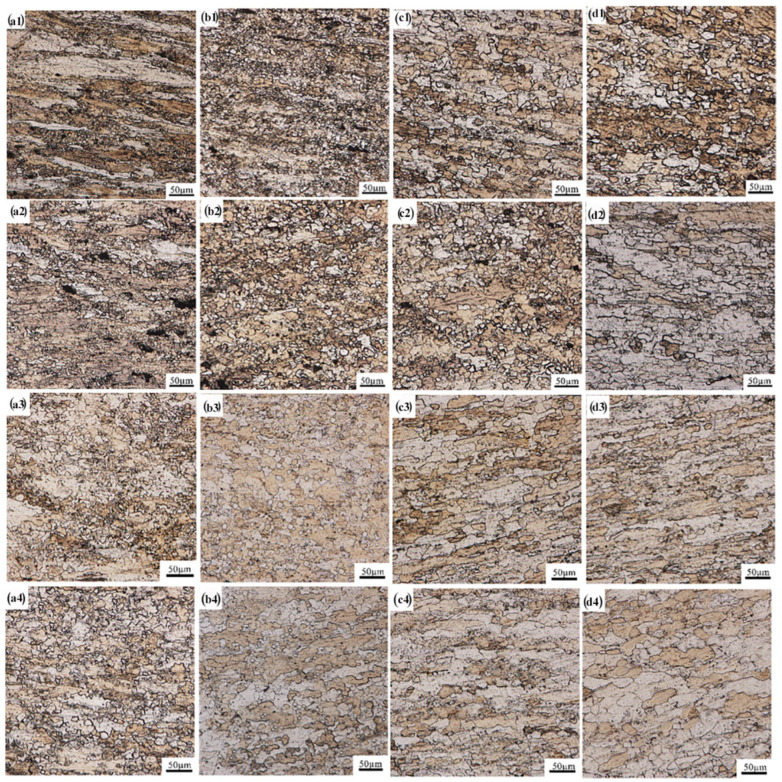
Microstructures of the gauge section of GAZ211 alloy at different deformation temperatures of 573 K (**a1**–**a4**), 623 K (**b1**–**b4**), 673 K (**c1**–**c4**), and 723 K (**d1**–**d4**) and strain rates of 1.67 × 10^−2^ s^−1^ (**a1**–**d1**), 1.67 × 10^−3^ s^−1^ (**a2**–**d2**), 5 × 10^−4^ s^−1^ (**a3**–**d3**), and 1.67 × 10^−4^ s^−1^ (**a4**–**d4**).

**Figure 5 materials-16-01639-f005:**
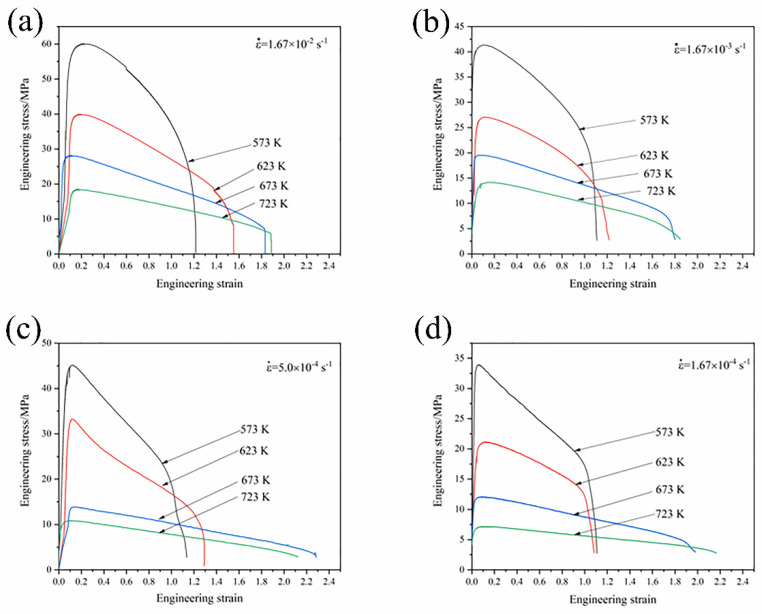
Engineering stress–strain curves of GAZ211 alloy at different temperatures and initial strain rates: (**a**) 1.67 × 10^−2^ s^−1^; (**b**) 1.67 × 10^−3^ s^−1^; (**c**) 5.0 × 10^−4^ s^−1^; (**d**) 1.67 × 10^−4^ s^−1^.

**Figure 6 materials-16-01639-f006:**
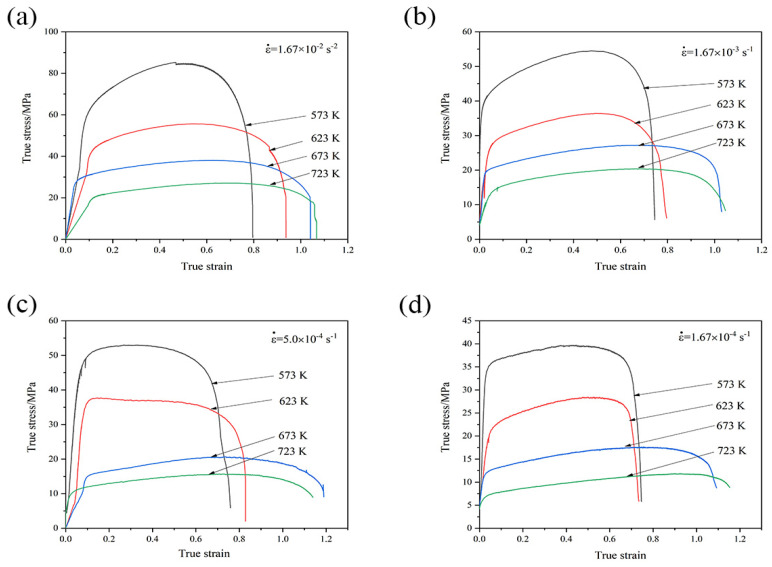
True stress–strain curves of GAZ211 alloy at different temperatures and initial strain rates: (**a**) 1.67 × 10^−2^ s^−1^; (**b**) 1.67 × 10^−3^ s^−1^; (**c**) 5.0 × 10^−4^ s^−1^; (**d**) 1.67 × 10^−4^ s^−1^.

**Figure 7 materials-16-01639-f007:**
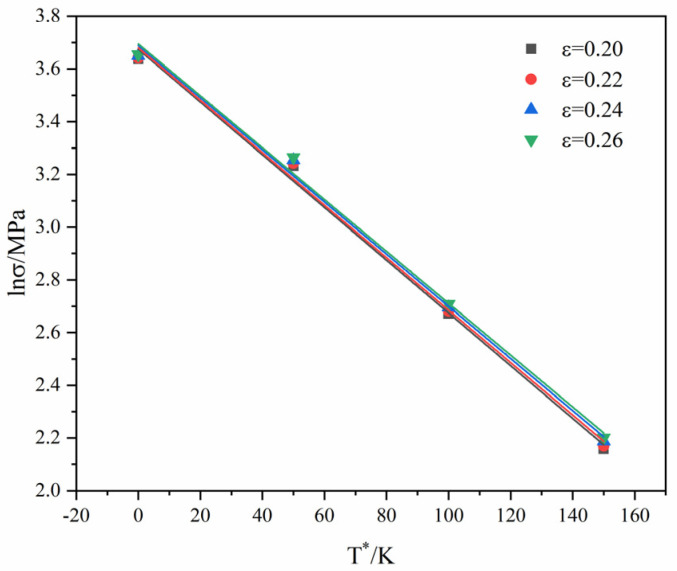
Linear fitting of lnσ against $T^{*}$.

**Figure 8 materials-16-01639-f008:**
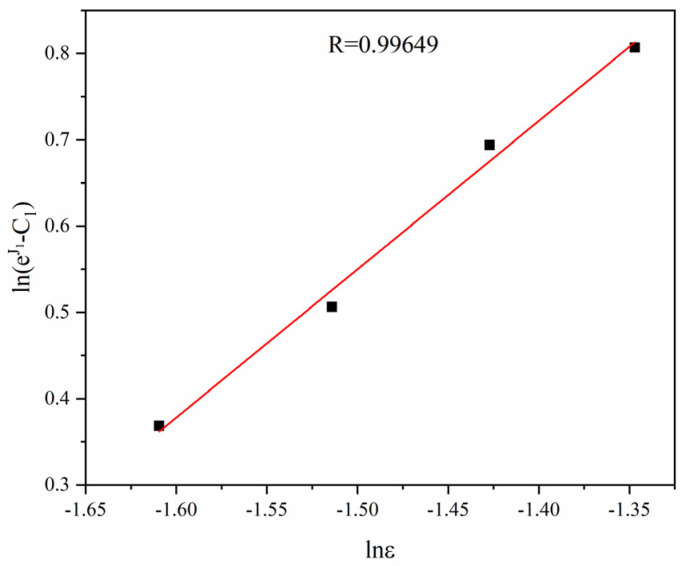
Linear fitting of $\ln\left(\exp J_{1}-C_{1}\right)\;$against lnε.

**Figure 9 materials-16-01639-f009:**
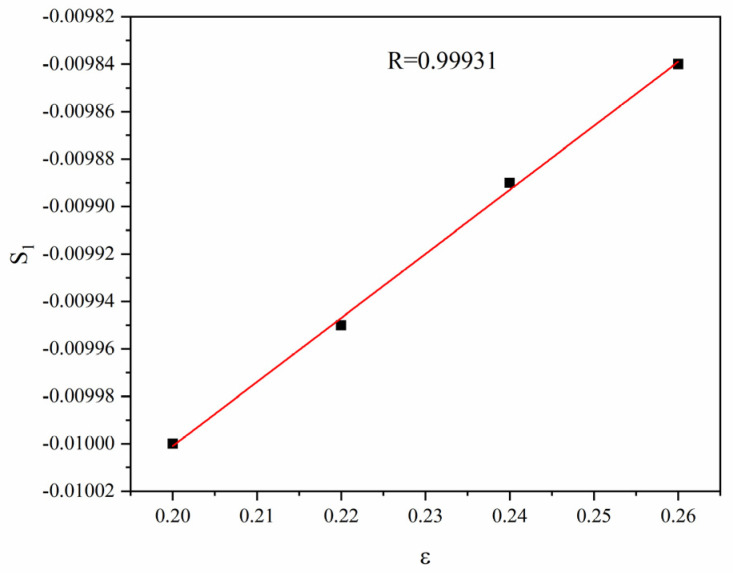
Linear fitting of $S_{1}\;$against ε.

**Figure 10 materials-16-01639-f010:**
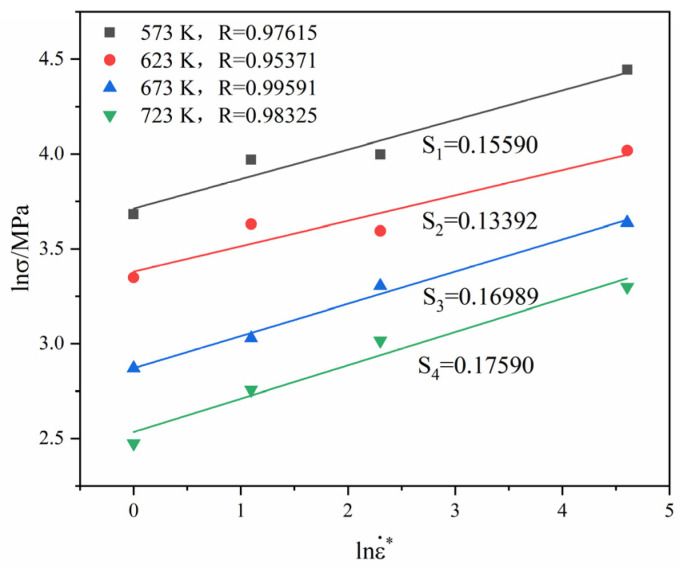
Linear fitting of lnσ against$\;\ln{\dot{\varepsilon }}^{*}$.

**Figure 11 materials-16-01639-f011:**
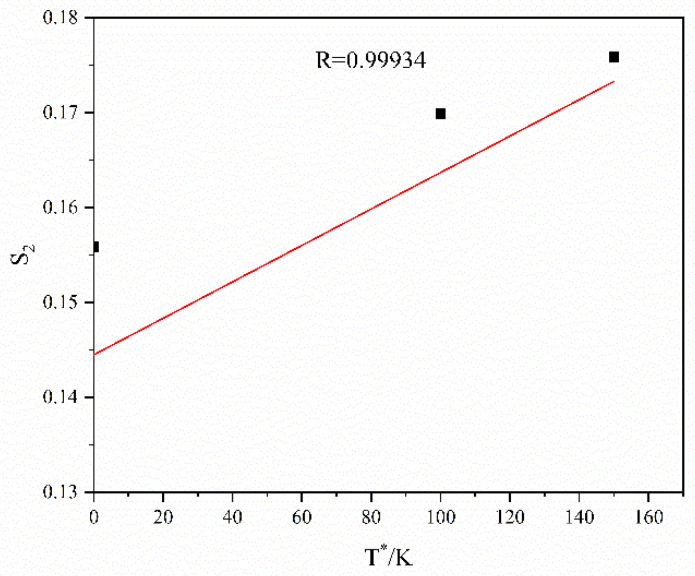
Linear fitting of $S_{2}\;$against$\;T^{*}$.

**Figure 12 materials-16-01639-f012:**
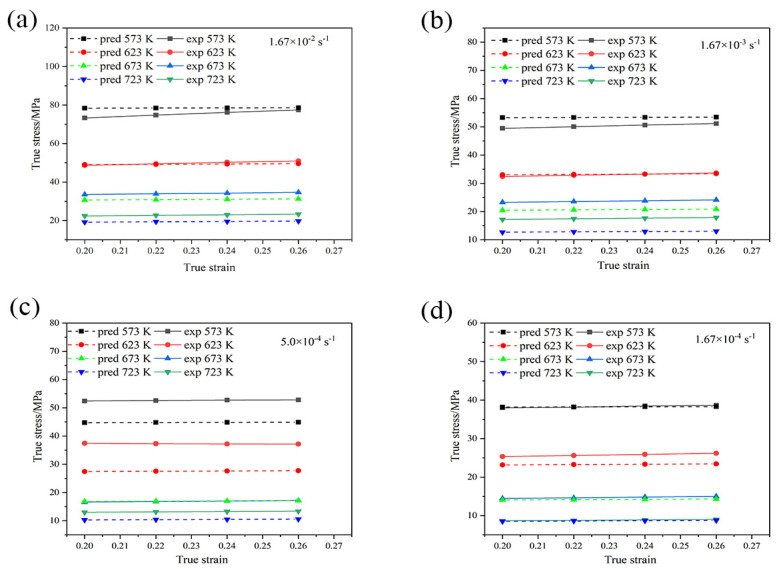
Comparison between experiment and predicted flow stress values with the modified Z-A model with different strain rates: (**a**) 1.67 × 10^−2^ s^−1^; (**b**) 1.67 × 10^−3^ s^−1^; (**c**) 5.0 × 10^−4^ s^−1^; (**d**) 1.67 × 10^−4^ s^−1^. pred indicates the calculated stress, and exp indicates the measured stress.

**Figure 13 materials-16-01639-f013:**
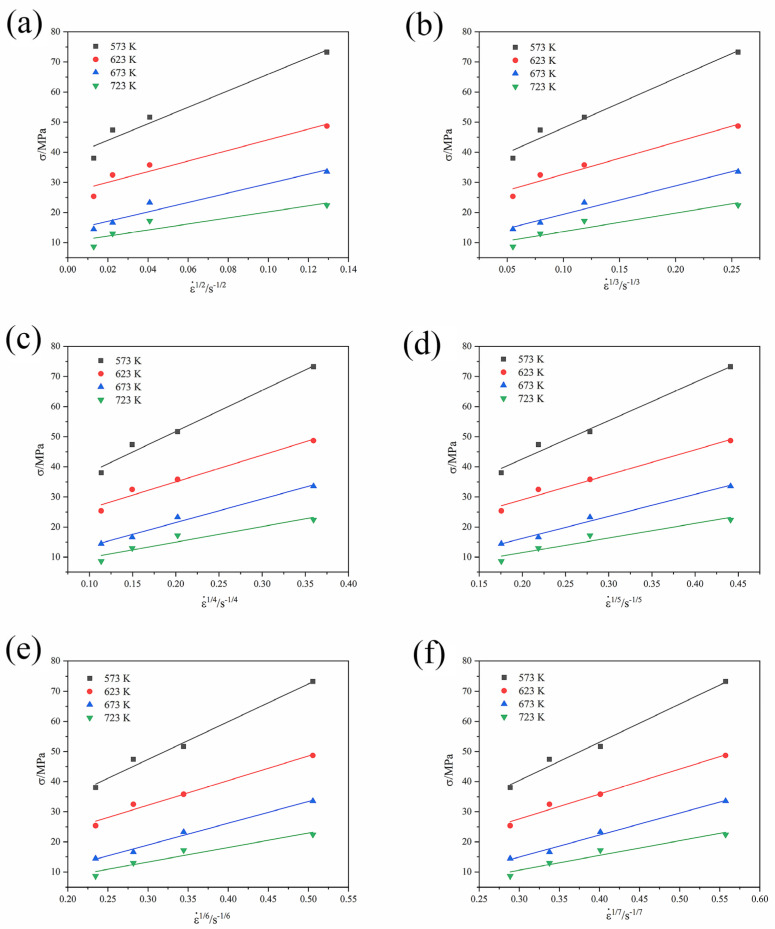
Linear fitting of *σ* against $\dot{\varepsilon }$^1/*n*^ for the GAZ211 alloy: (**a**) *n* = 2; (**b**) *n* = 3; (**c**) *n* = 4; (**d**) *n* = 5; (**e**) *n* = 6; (**f**) *n* = 7.

**Figure 14 materials-16-01639-f014:**
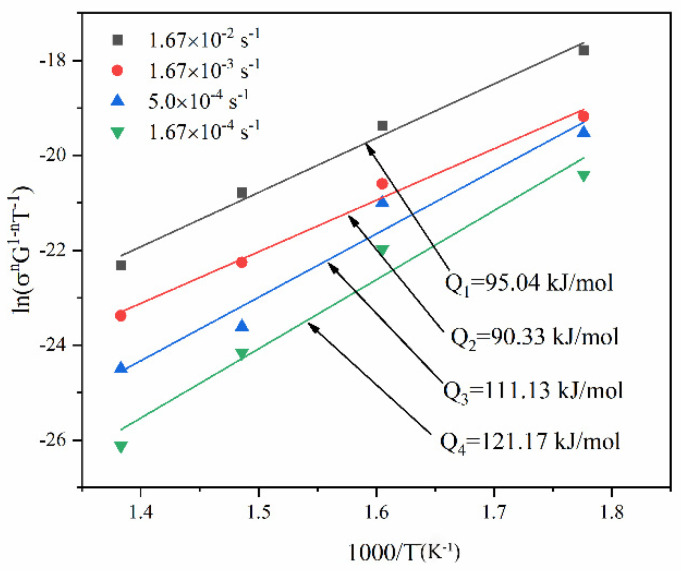
Fitting curves of $\ln\left(\sigma^{n}G^{1-n}T^{-1}\right)-1/T$ in GAZ211 alloy at different strain rates.

**Figure 15 materials-16-01639-f015:**
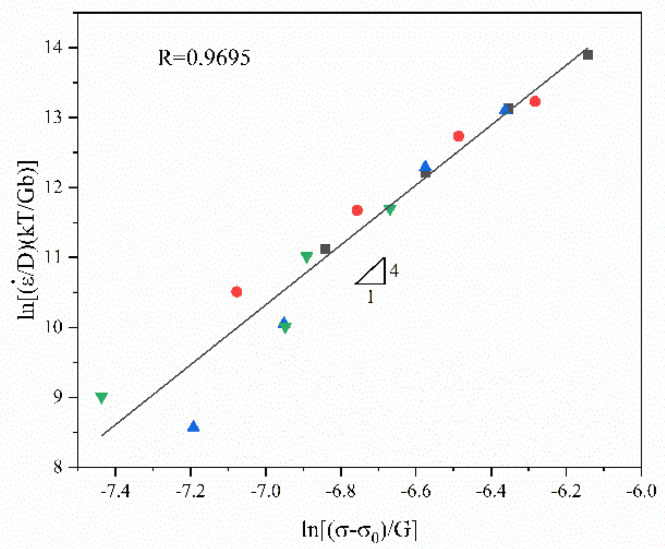
Normalized curve of GAZ211 alloy. The data points are normalized experimental data.

**Figure 16 materials-16-01639-f016:**
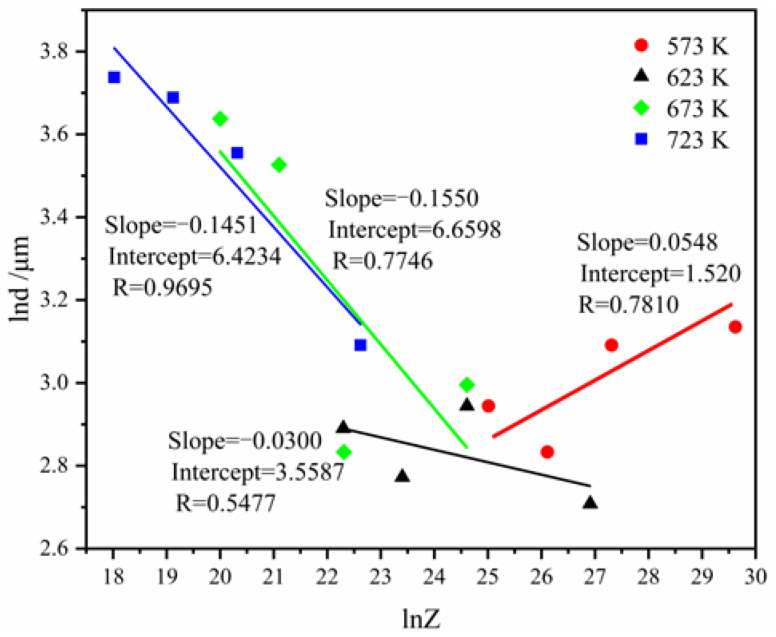
The plot of ln*d* as a function of ln*Z*.

**Table 1 materials-16-01639-t001:** Grain sizes and elongations of GAZ211 alloy tensiled at different temperatures and strain rates.

Temperature/K	Strain Rate/s^−1^	Grain Size/μm	Elongation/%
573 K	1.67 × 10^−2^	23 ± 4.1	121.6
	1.67 × 10^−3^	22 ± 2.0	110.8
	5 × 10^−4^	17 ± 2.7	113.6
	1.67 × 10^−4^	19 ± 3.2	110.9
623 K	1.67 × 10^−2^	15 ± 1.5	155.3
	1.67 × 10^−3^	19 ± 3.1	121.6
	5 × 10^−4^	16 ± 2.8	129.1
	1.67 × 10^−4^	18 ± 2.4	108.3
673 K	1.67 × 10^−2^	20 ± 3.0	189.1
	1.67 × 10^−3^	17 ± 1.8	180.1
	5 × 10^−4^	34 ± 3.2	228.4
	1.67 × 10^−4^	38 ± 3.4	198.2
723 K	1.67 × 10^−2^	22 ± 2.9	190.7
	1.67 × 10^−3^	35 ± 3.5	184.6
	5 × 10^−4^	40 ± 4.0	212.4
	1.67 × 10^−4^	42 ± 2.7	216.6

**Table 2 materials-16-01639-t002:** Parameters for the modified Zerilli–Armstrong model.

Parameter	*C*_1_/MPa	*C*_2_/MPa	*n*	*C* _3_	*C* _4_	*C* _5_	*C* _6_
Value	38.00	3.13	1.71943	0.01054	−0.0027	0.15604	0.00013

## Data Availability

The data presented in this study are available on request from the corresponding author.

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
