# Peer review of "Room-Temperature Strengthening, Portevin-Le Chatelier Effect, High-Temperature Tensile Deformation Behavior, and Constitutive Modeling in a Lightweight Mg-Gd-Al-Zn Alloy"

_materials, 2023, doi:10.3390/ma16041639_

Round 1

Reviewer 1 Report

The manuscript requires a major revision. Specific comments are listed as follows:

1. Introduction – the authors wanted to include everything, whilst the motivation to design such a Mg-2Gd-1Al-1Zn alloy as well as study its deformation behavior is poorly clarified. Of course, given such a specific composition, it lacks significant researches in the literature on studying the strengthening mechanism at room temperature and creep mechanism at high temperatures. Therefore, it should be clarified why such a composition is designed ? How it is different from other Mg-Gd-Zn or Mg-Gd-Al-Zn alloys in the literature?

Line 58-65, I do not see any point in discussing strengthening mechanisms in Al alloys as well as the Portevin-Le Chatelier effect in other alloys with the relation to the Mg alloys studied here. These text should be largely sharpened.

Line 72-77, can you be more specific and focused on what you studied with the Mg alloys at high temperature ? The text suggest you studied everything.

2. Figure 1. Text for the scale bars should have the same size; please indicate the rolling direction in (c).

3. Line 313, what is the melting temperature of this Mg-2Gd-1Al-1Zn alloy ?

4. Line 437, how you determine the threshold stress ? Is it a constant value at all temperatures ?

5. Line 428-449, in the Sherby-Dorn equation as expressed by Eq. 12, is the stress exponent n required to be an integer as you presented as possible values of 2-7 ? I do not think so. I suggest you fit the data with the best goodness to extract the value for n. Therefore, the whole analysis here should be redone.

6. Line 457, can you really say in this condition it is a creep process ? I would think it is a high temperature tensile testing / deformation, not a slow creep deformation process.

7. Figure 14 and Line 463, How do the values of these activation energies compared to the activation energy of Mg self diffusion ? This should be discussed in detail in order to better understand how dislocation and diffusion of vacancies dominates / interplays against each other ?

8. Line 464-478, I could not understand why the slope, expressed by n, is derived as 4.2 from Figure 15, whereas it is expressed as a value of 3 for the stress exponent in Eq. 15. Can you clarify this ?

9. Section 4.1, the calculation of different strengthening contributions rely heavily on rough estimation of many parameters. Therefore, you should point out and emphasize that the calculation here only gives very rough estimation on the fractions of different contributions. For more quantitative analysis, one would require better or more detailed microstructural characterization to derive the relevant parameters.

10.  Section 4.5 and 4.6 largely overlapped with each other. Maybe it is better to be merged and be more concise.

11. Concerning the microstructure change during the high temperature deformation, except for the dynamic recrystallization, do you also observe cavity nucleated in/at/across the second phase particles ? Since the elongation is so big, I would imagine that some cavities have formed before fracture. Have you looked into the locations of these cavities ? Please include these observations into the manuscript.

12. Text flaws:

Line 47, second phase => precipitation

Line 47, “and”;

Reviewer 2 Report

This article is very interesting and present in excellent form the results of the research. This topic is in the main stream of the research of novel light materials for the industry and production technologies, which improve the strength properties. All information are presented clearly, methods are described in details and obtained results are commented, explained and assessed.

Below are pointed some errors or changes which can improve this article:

#1 line 6 – there is not affiliation no. 3 indicated in line 9

#2 line 47 – “… the second phase strengthening and. Meanwhile, The purpose of…” correct “and” and “The” in this sentence/sentences

#3 lines 47, 104, 113, 132 and others places – at the end of line are the values, but its unit are in the next line. This must be corrected in whole article

#4 lines 72-74 – in this sentence three times is “such as” – please correct this sentence

#5 lines 109, 123, 125 – the dimensions should be in mm, not “mm3” – it is a volume

#6 Figure 1 – there are two different scales

#7 line 173 – capital letter “E” should be corrected

#8 references in the text concerning figures and equations in the article is wrong – e.g. “Figure 1”, not “Figure.1” (with a dot)

#9 there are three different signs of multiplication e.g. lines 242 (bold), 253, 282

#10 Figure 5 – there are individual, different scales on both axes – there will be better if the scale will be the same

#11 Figure 11 – there is temperature -20K (!!!) – delete the range below 0K

#12 line 410 – “Since we only established…” – change from the personal to the impersonal form/passive voice

#13 Figure 13 – the points (a), (b) … are to large

#14 lines 485-487 – sign “n1” should be rather “n1” in all cases

#15 line 512 – lack of subpoint (1)

#16 line 696 – “… strain rate, The modelling…” – correct this place in sentence/sentences

#17 point 4.6 and 5. – the information in point 4.6 is multiplied in conclusion. Point 4.6 must be also rearranged. In the sentences started at lines 725 and 735 are information concerning “bullet points”. Information below these sentences are some individual points and thus should be pointed.

Reviewer 3 Report

In my opinion, well written research. Some of the suggestions for improvement are as follows:

1. I don't think group citations are necessary. For example 1-7, etc.

2. Errors should be analyzed and discussed. Additionally, it would be good to estimate the measurement uncertainty.

3. Subsection 4.6. A novelty statement in this manuscript is not necessary at this point in the manuscript. Part of this section should be at the end of the Introduction section and part in the Conclusions section.

4. The Conclusions section should be corrected. Results should not be repeated. Highlight only the most important. In the Conclusions section, the scientific contribution should be highlighted. Possibilities of practical application should also be mentioned. Finally, limitations and future research should be written.

Round 2

Reviewer 1 Report

Additional comments are presented below:

1. The problem still exists in Figure 1. Different text font sizes and no indication for the the rolling direction in (c).

2. Line 177-216, How these newly added standard errors for the metrics of the mechanical properties derived ? This should be clarified in the manuscript.

3. Table 1. For the standard deviations of the grain sizes, why some of them have one digit, being different from the others ?

4. So many values are added behind “±”, how much we can trust those values ?

5. The authors cite several papers from Ruano, Wadsworth and Sherby, in particular for the integer values of exponent n. However, the authors also use Equation 12 to describe and fit their experimental data, whilst in Equation 12 there is a threshold stress, whose existence is largely questioned by Ruano, Wadsworth and Sherby (e.g. Deformation mechanisms in an austenitic stainless steel (25Cr-20Ni) at elevated temperature, J. Mater. Sci, 1985, 3735-3744). So what do you think you can unify these contradictory phenomenongical equations together ? Should the values for the stress exponent really be integer numbers and corresponding to so specific deformation mechanism ? No coupled effects ?

6. Line 653-656, the text is contradictory. Why “the cavity morphology is observed” and then “Hence, the cavity development is not obvious”.
